# Impact of protein identity on tumor-associated antigen uptake into infiltrating immune cells: A comparison of different fluorescent proteins as model antigens

Rulan Yi[1], Emily Chen[2], Edward W. Roberts[3], Matthew F. Krummel[2]*, Nina Kathrin Serwas[2]*

1 MCB Undergraduate Program, University of California, Berkeley, CA, United States of America, 2 Department of Pathology, University of California, San Francisco, CA, United States of America, 3 Cancer Research UK Beatson Institute, Glasgow, United Kingdom

* matthew.krummel@ucsf.edu (MFK); nina.serwas@ucsf.edu (NKS)

**Data Availability Statement:** All relevant data are within the paper and its Supporting Information files.

## Abstract

Effective immune responses depend on efficient antigen uptake in the periphery, transport of those antigens to, and presentation in draining lymph nodes (LNs). These processes have been studied intensively using stable fluorescent proteins (FPs) as model antigens. To date, ZsGreen is the only FP that can be tracked efficiently towards LNs, hence, it is difficult to compare studies using alternated tracking proteins. Here, we systematically compared six different FPs. We included ZsGreen, ZsYellow, DsRed, AsRed, mCherry, and mRFP based on sequence homology and/or origin species, and generated FP-expressing tumor cell lines. Stability of fluorescent signal was assessed *in vitro* over time, across different pH environments, and *in vivo* through FP antigen uptake and transfer to immune cells isolated from tumors and tumor-draining LNs. ZsGreen could be detected in high percentages of all analyzed tumor-infiltrating immune cells, with highest amounts in tumor-associated macrophages (TAMs) and type 2 conventional dendritic cells (cDC2s). ZsYellow, AsRed, and DsRed followed a similar pattern, but percentages of FP-containing immune cells in the tumor were lower than for ZsGreen. Strikingly, mRFP and mCherry demonstrated a 'non-canonical' antigen uptake pattern where percentages of FP-positive tumor-infiltrating immune cells were highest for cDC1s not TAMs and cDC2s despite comparable stabilities and localization of all FPs. Analysis of antigen-containing cells in the LN was hindered by intracellular degradation of FPs. Only ZsGreen could be efficiently tracked to the LN, though some signal was measurable for ZsYellow and DsRed. In summary, we find that detection of antigen uptake and distribution is subject to variabilities related to fluorophore nature. Future experiments need to consider that these processes might be impacted by protein expression, stability, or other unknown factors. Thus, our data sheds light on potential under-appreciated mechanisms regulating antigen transfer and highlights potential uses and necessary caveats to interpretation based on FP use.

**Funding:** This work was supported by Human Frontier Science Program in the form of a fellowship (LT000061/2018-L) to NKS. This work was also supported by National Institutes of Health in the form of a grant (5U01CA217864) to MFK. The funders had no role in study design, data collection and analysis, decision to publish, or preparation of the manuscript.

**Competing interests:** The authors have declared that no competing interests exist.

## Introduction

For a successful tumor immune response, the proper uptake and presentation of tumor-associated antigens (TAAs) by antigen presenting cells (APCs), such as dendritic cells (DCs), is essential. As such, antigens from peripheral tissues need to be taken up by phagocytic APCs which mature, upregulate costimulatory molecules, and present these antigens as processed peptides on MHC complexes. During this process, APCs traffic from the peripheral tissue where the tumor is located to the tumor-draining lymph node (tdLN). There, the APCs can interact with T and B cells to generate an effective long-term response [1, 2]. The interaction of the TAA-peptide:MHC complex and its accessory proteins with receptors on T and B cells determines downstream immune responses such as cell activation and differentiation. Exactly how TAAs are taken up and brought to the LN for presentation is not fully understood.

Previous studies showed that type 1 DCs (cDC1s) are essential for the transport of TAAs to the lymph node, and that this transport is dependent on C-C chemokine receptor 7 (Ccr7) [3, 4]. Upon entry into the LN, migratory DCs can transfer antigens to other lymph node resident DCs as membrane encapsulated vesicles through direct cell-cell contact [5]. The choice of the receiving DCs may then dictate the downstream immune responses generated through subsequent T cell interaction causing activation, proliferation, memory induction, and/or effector responses in the T cell [5]. Previous studies that analyzed the process of antigen uptake, presentation, and transfer have used fluorescent proteins (FPs) such as mCherry, green fluorescent protein (GFP), yellow fluorescent protein (YFP), tdTomato, and ZsGreen as model antigens [3–5]. FPs used in these studies seemed to have different stabilities since only ZsGreen was substantially detected in resident dendritic cells. Importantly, other tumor antigens can be tracked in those resident DCs as well, confirming that ZsGreen presence was not an artifact, but faithfully reported on transport [5]. It is not clear why other FPs cannot be transported to the same extend to the tdLNs and whether certain properties of FPs alter antigen uptake and distribution or simply degrade more rapidly than others *en route* to the LN.

A wide range of FPs exists that are frequently used in imaging and tracking experiments. These FPs have varying brightness and stabilities that may potentially impact the outcome of antigen uptake, transfer, and presentation. Brightness is defined as extinction coefficient times quantum yield at pH 7.4 [6] and can range from dim proteins, such as AsRed derived from *Anemonia sulcate*, with a brightness index of 2.81 [7], to bright proteins with indices of 90 (tdTomato) [6]. Stability is dependent on the specific environment in which the FP resides at the time of analysis [6]. In this study we tested antigen stability *in vitro* and *in vivo* for six different FPs with a range of brightness indexes between 2.81 and 49. We tracked their stabilities during uptake and intracellular storage in tumor-infiltrating immune cells and after transport to the tdLNs. The combined results offer a comprehensive overview of the properties of the chosen FPs with regards for their usefulness in assays tracking antigen uptake and transfer. This work will facilitate the choice of FPs in studies that will use two or more fluorophores in parallel.

## Material and methods

### Alignment

Amino acid sequences of all fluorescent proteins used were gathered via FPbase [7], then aligned via Basic Local Alignment Search Tool (BLAST) [8]. Percent similarity to ZsGreen and DsRed were determined using the global align tool.

## Mice

Mice were housed and bred under specific pathogen free conditions at the University of California, San Francisco Laboratory Animal Research Center and all experiments conformed to ethical principles and guidelines approved by the UCSF Institutional Animal Care and Use Committee. C57BL/6 mice were purchased from Jackson Laboratory or bred in-house. Male mice between 8–12 weeks of age were used.

## Cell lines

Vectors containing target fluorescent genes ZsGreen, ZsYellow, DsRed, AsRed, mCherry, and mRFP (Takara Bio), were either transfected into B16-F10 cells using the calcium chloride precipitation method (all FPs except ZsGreen), or retrovirally transduced into B16F10 cells (ZsGreen). Transformed cells were then harvested, washed, and repeatedly sorted through fluorescent activated cell sorting (FACS) on a BD FACSAria™ to select for stable fluorescent positive populations. Sorted cells were then collected, frozen and/or propagated for use in future experiments. Cells were cultured at 37˚C with 5% $CO_2$ in DMEM (GIBCO) plus 10% heat-inactivated FCS with penicillin, streptomycin, and L-glutamate on tissue-culture treated plastic plates and split every other day.

## Fluorescence stability assay

FP-expressing B16-F10 cells were harvested with trypsin, counted, and washed in ice cold PBS. Then, cells were homogenized with mechanical force generated through repeated passage of cells through a 22-gauge needle in hypotonic lysis buffer (20mM HEPES buffer, 2mM EGTA, 2mM MgCl2). Lysates were transferred to 96-well black microplates suitable for fluorescent based assays (Invitrogen). Each well contained lysates of 1 Mio cells. pH was adjusted with NaOH or HCl, controlled with a pH-meter before the assay, and controlled with pH-indicator strips on the lysates after the experiment to ensure that the pH was stable overtime. For each different fluorophore and pH condition, three technical replicates were performed. Data was collected over varying time points using a microplate reader (SpectraMax) and incubated at 37˚C or 4C, respectively. Analysis of whole cells was done similarly in separate plates. The experiment was done in biological triplicates.

## Ectopic tumor injections

B16-F10 cells expressing the respective FP were grown to 80% confluency, then harvested, washed, and counted. Cells were resuspended in a suitable amount of PBS and mixed at a 1:1 (v:v) ratio with growth factor reduced Matrigel Matrix (BD Biosciences). A final injection volume of 100ul containing 400k cells was injected subcutaneously on the flanks of mice.

## Harvest, collection of data, and analysis

Tumors, inguinal and axillary lymph nodes were harvested 2 weeks after injection. After digestion with DNAse, Collagenase I and Collagenase IV, washing, and counting, 10 million cells per FP were stained with fixable Live/Dead Zombie (BioLegend) and respective antibodies: MHCII-BV421, F4/80-BV510, CD11b-BV605, CD11c-BV650, Ly6C-BV711, CD45R-BV785, CD90.2-BV785, Ly6G-BV785, NK1.1-BV785, CD8alpha-PerCPCy5.5, CD103-APC, CD45-AF700, CD24-PECy7 (Biolegend, eBioscience).

Flow cytometry was performed on a BD LSR Fortessa instrument at the ImmunoX Flow Cytometry CoLab. Analysis of flow cytometry data was done using FlowJo (Treestar) software,

and graphs and figures were produced with GraphPad Prism software. All experiments were done in triplicates.

### Fluorescence microscopy

Cells from each respective fluorescent cell line were thawed and cultured. 50,000 cells were seeded in a poly-L-lysine coated 8-well chambered coverslip and let rest overnight. For live cell imaging, a Leica SP5 laser scanning confocal microscope with an incubation chamber and a 20X oil objective was used to visualize adherent cells expressing respectively ZsGreen, ZsYellow, AsRed, DsRed, mRFP, and mCherry. The experiment was done in triplicates.

## Results and discussion

### Characteristics of selected fluorescent protein

We compared five different fluorescent proteins (FPs) for antigen uptake and transfer characteristics to ZsGreen. ZsGreen is to date the only fluorophore that can be efficiently tracked from expressing tumors to lymph node resident dendritic cells. Our selection (Table 1) was based on species origin (ZsYellow), brightness of the fluorophore (DsRed), sequence homology (mCherry, mRFP) and availability of corresponding biochemical tools (AsRed, mRFP, mCherry). ZsGreen, is derived from corals of the *Zoanthus sp*, and has been shown to have high stability in various chemical environments [9]. Its brightness index is 22.43 [7]. ZsYellow is a stable protein purified from the same species as ZsGreen, and is excited at a wavelength around 528 nm with a brightness index of 13 [7, 9]. DsRed is derived from *Discosoma sp* [10], excited by a wavelength of 558 nm, and has a brightness index of 49.3 [7]. Naturally occurring FPs such as DsRed have been modified to achieve proteins with optimized brightness and stability properties such as mRFP and mCherry, with brightness indices of 12.5 and 15.84 respectively [7]. In addition, these modified proteins exist as monomers in contrast to the tetrameric nature of ZsGreen and DsRed [7]. AsRed has the lowest brightness in our collection (2.81) and is expressed in the coral species *Anemonia sulcate* [6, 7, 10].

FP-encoding vectors were transfected or transduced into B16-F10 cell lines. Transformed cells with the highest signal for the respective FP were repeatedly sorted with a 5%- cut-off using fluorescence activated cell sorting (FACS) (Fig 1A and 1B). All generated cell lines reached high FP positivity with a 2–3 log fluorescent intensity difference over un-transformed cells (Fig 1A and 1B). Despite repeated and stringent selection for FP-positivity, both ZsYellow- and mCherry-transfected B16-F10 cells kept a substantial amount of non-fluorescent cells in their cultures (Fig 1B). This might be due to non-stable integration or epigenetic silencing of the transgene. In subsequent analyses we considered this bimodal distribution as a potential factor for differences in antigen uptake or distribution. Previous work has shown

**Table 1. Fluorophore properties [7, 8].**

| Fluorophore | Brightness | Excitation | Emission | % similarity to ZsGreen | % similarity to DsRed |
|---|---|---|---|---|---|
| AsRed | 2.81 | 576 nm | 592 nm | 53.78% | 61.54% |
| mRFP | 12.5 | 584 nm | 607 nm | 56.65% | 87.56% |
| ZsYellow | 13 | 528 nm | 539 nm | 91.77% | 58.87% |
| mCherry | 15.84 | 587 nm | 610 nm | 53.53% | 82.63% |
| ZsGreen | 22.43 | 496 nm | 506 nm | — | 59.23% |
| DsRed | 49.3 | 558 nm | 583 nm | 59.23% | — |

Summary of used fluorophores and their respective similarity with ZsGreen and DsRed [7, 8]

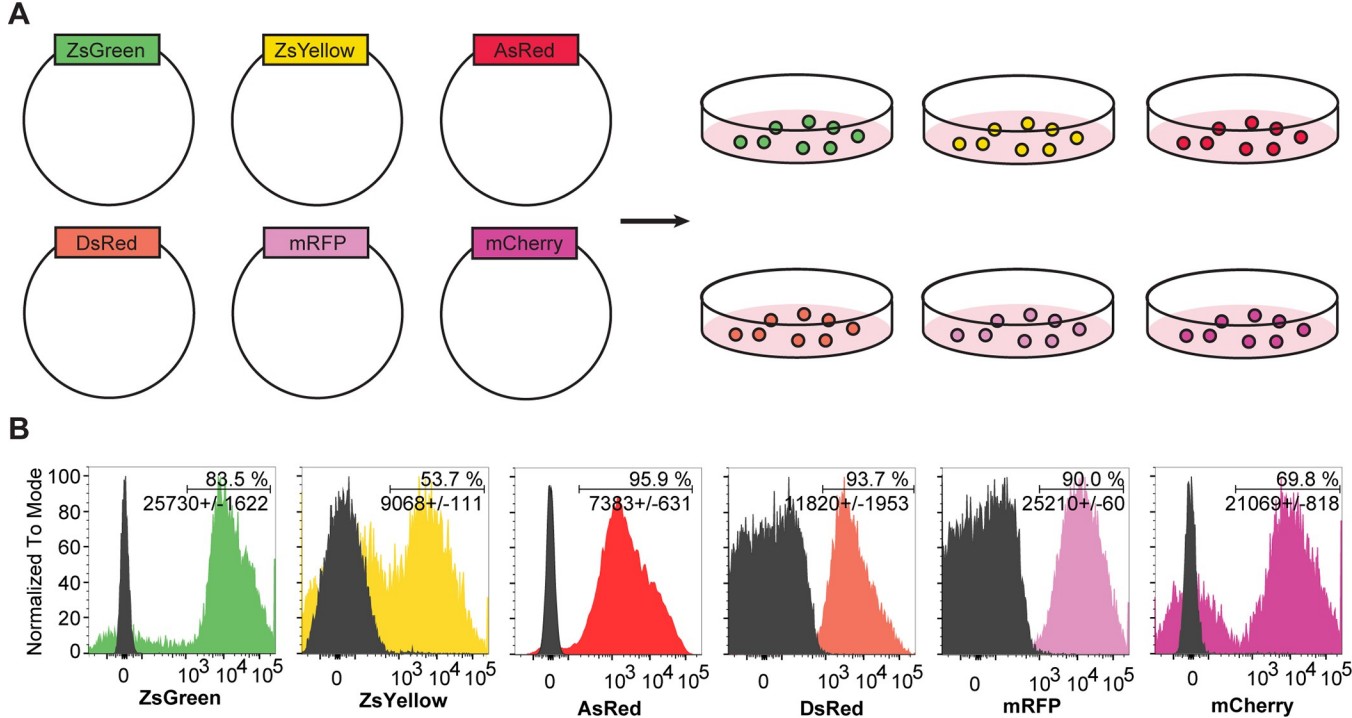

**Fig 1. Generation of cell lines with respective fluorescent genes for downstream analysis. A)** Retroviral vectors expressing respective fluorescent genes ZsGreen, ZsYellow, mCherry, mRFP, DsRed, and AsRed were transfected/transduced into B16-10 cells. **B)** Fluorescence-activated cell sorting (FACS) was then used to select for FP positivity. ZsYellow and mCherry cells display a bimodal pattern despite undergoing FACS multiple times with a selection criterion for the top 5% of FP positive cells. Histograms are shown with % positive cells and average mean fluorescence intensity of the chosen fluorophore (+/- standard deviation).

that differential subcellular localization has an effect on antigen uptake and distribution into compartments of the immune system [11]. All generated cell lines had their respective FPs evenly distributed in the cytoplasm (S1 Fig).

## Assessment of *in vitro* stability

The stability of fluorophores expressed in the generated cell lines was assessed in an *in vitro* fluorescent stability assay. For this purpose, FP-expressing B16-F10 cell lines were lysed in a hypotonic lysis buffer. Lysates were then kept at 37°C to mimic the cellular environment. No protease inhibitors were added. This allowed us to follow protein stability in the cytosolic soup containing various proteases and other enzymes, through fluorescence intensity, over time. As this assay investigates the stability of FPs, and not their individual brightness, we internally normalized the brightness to timepoint 0 in neutral pH conditions. In general, ZsGreen demonstrated the highest stability of all analyzed FPs, and showed only a minimal decrease in fluorescent signal over the time frame of 6 hours (Fig 2A). This high stability of ZsGreen is in concordance to previous published work [9]. In contrast, all other FPs showed a severe reduction of signal over the first two hours (Fig 2A). Acidification of the protein lysates showed relative robustness of all FPs to pH alterations (Fig 2B). The highest fluorescent intensity of all FPs could be detected at neutral pH conditions which is in line with previous published analyses of pH stability for fluorescent proteins [12]. Our data suggests that the impact of acidification in certain intracellular compartments only plays a minimal role on FP fluorescence intensity and stability. However, proteins might be degraded over time through intracellular proteases or

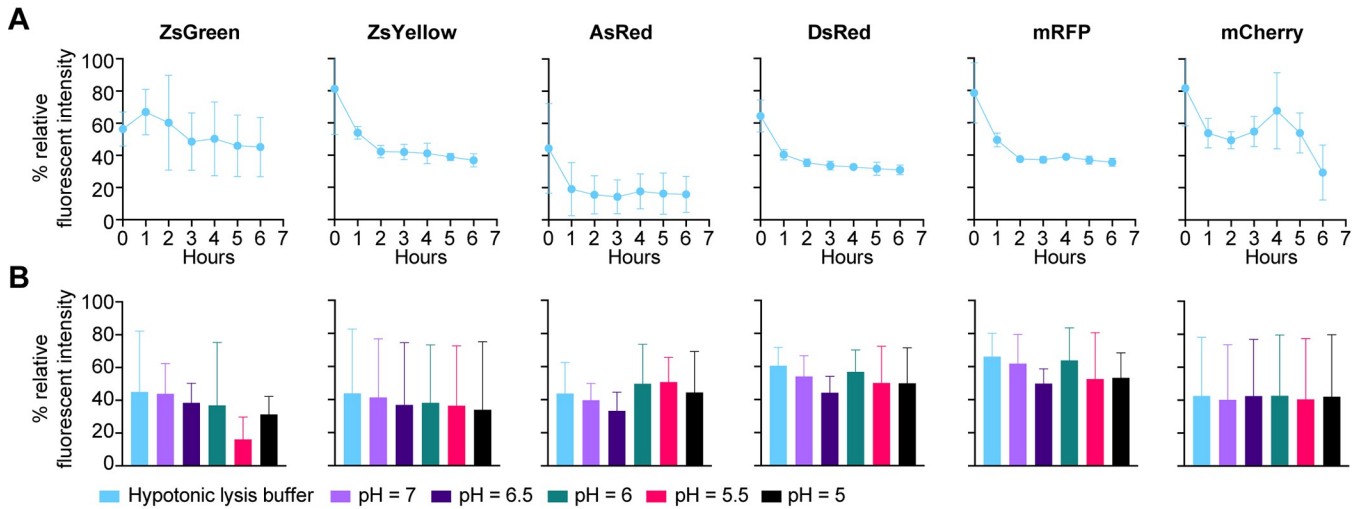

**Fig 2. *In vitro* stability assay shows comparable stabilities of selected fluorophores.** Lysates of 1 million respective FP-expressing B16-F10 cells were generated in hypotonic lysis buffer and incubated at **A)** 37˚C in pH 7.4 or **B)** 37˚C in pH ranging from 5–7.4. Fluorescence intensity data was collected over time at the indicated time points (**A**) or immediately following pH treatment (**B**). Data was normalized to the maximum fluorescence intensity of each FP for every independent experiment.

other enzymes, which was especially true for ZsYellow, AsRed, DsRed, mRFP, and mCherry. ZsGreen markedly showed resistance towards degradation by proteins contained in the cellular lysates which might explain its superior performance in tumor-associated antigen tracking experiments. Differences between FPs other than ZsGreen in those assays seem to not be caused by different fluorescence stabilities.

## Tumor-associated antigen uptake varies for different FPs

To evaluate tumor-associated antigen (TAA) uptake of the six selected FPs, cells were injected bilaterally into the flanks of B6 mice (Fig 3A). Tumors averaged a size of 1300 mm³ for all FP-containing cell lines two weeks after inocculation (Fig 3B). None of the generated FP-expressing B16-F10 cell lines resulted in severe increase or reduction of tumor growth, suggesting that the modification did not alter proliferation or fitness of the cells. For antigen tracking analysis, only tumors of a size equal to or greater than 800 mm³ were included.

We assessed distribution of FPs into different cellular populations of the tumor, namely monocytes, neutrophils, tumor associated macrophages (TAMs), and DCs (Fig 4 and S2 Fig).

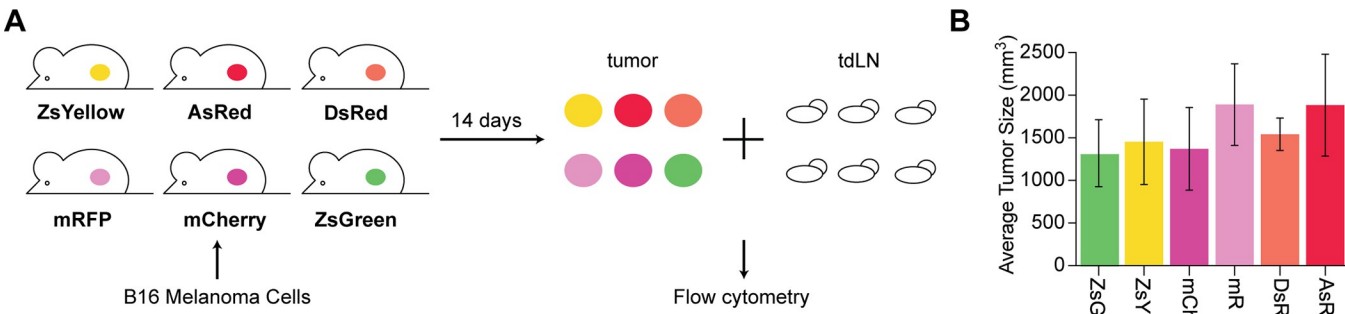

**Fig 3. Growth and harvest of respective fluorescent cell lines. A)** Fluorescent cell lines were grown, harvested, and injected subcutaneously on the flanks of B6 mice. After 14 days, tumors and draining lymph nodes were harvested and processed for flow cytometry. **B)** Average sizes of tumors (mm³) at harvest are shown for all six respective fluorophores. Size was controlled to ensure comparable *in vivo* growth of different FP-expressing B16-F10 cell lines.

Of note, infiltrating immune cell composition were similar for all injected tumors (S3 Fig). Among the selected FPs, percentages of FP-containing immune cells were highest in tumors expressing ZsGreen. Interestingly, we identified two distinct antigen uptake pattern: The 'classical' pattern showed highest percentages of FP-containing cells for cDC2s, TAM1s, and TAM2s. ZsGreen-expressing tumor cells resulted in nearly 100% of these infiltrating immune cells to contain the FP. ZsYellow-, DsRed- and AsRed-expressing tumor cells resulted in less FP-positive immune cells, however, cDC2s, and TAMs were still the prime samplers of tumor associated FPs. For this 'classical' uptake pattern, a smaller fraction of neutrophils, monocytes, and cDC1s displayed FP signal than seen for cDC2s and TAMs. The 'non-canonical' antigen uptake pattern was characterized by highest proportions of FP-positive cells for cDC1s and not cDC2s or TAMs. Also for this pattern, the lowest amount of transferred antigen could be seen in neutrophils and monocytes. Both, mRFP- and mCherry-expressing tumors resulted in such a distinct pattern of FP uptake by immune cells (Fig 4). Despite comparable brightness, mCherry resulted in overall higher percentages of FP-positive tumor-infiltrating immune cells than mRFP (Fig 4). Remarkably, this was true although mCherry-expressing cells were a mixture of only two thirds transformed and one third WT cells (Fig 1B).

Next, we analyzed the average amount of FPs within each of the individual populations. We used normalized delta mean fluorescence intensity to visualize how much sampled FP each cell population contained in comparison to other immune cells sampling from the same fluorophore. Overall patterns of relative uptake per cell (Fig 5) showed more similarity than the percentages of FP- positive cells (Fig 4). For most of the tumors, cDC2s and TAMs were the cells that contained most FP. The only exception from this was mRFP containing neutrophils which on average contained more FP than TAM2s. Despite showing a 'non-canonical' uptake pattern, cDC1s infiltrating mRFP and mCherry-expressing tumors did not contain an increased amount of antigen.

It remains unclear what causes the increased percentage of FP-positive tumor infiltrating cDC1s in mRFP and mCherry-expressing tumors. Numbers of cDC1s are not different between tumors expressing FPs irrespectively of whether they cause 'classical' or 'non-canonical' antigen uptake patterns (S3 Fig). Of note, whereas ZsGreen, ZsYellow, AsRed, and DsRed are proteins directly isolated from distinct organism, mCherry and mRFP have been designed to better serve as tools for molecular biology. Changes aimed to enhance brightness and stability, and to prevent dimerization [10, 13].

## ZsGreen, ZsYellow, and DsRed can be tracked in the tdLN

All investigated FPs could be detected in tumor-infiltrating immune cells to a certain extend when expressed in injected B16-F10 tumor cells. Next, we wanted to determine whether selected fluorophores remained intact after trafficking of the containing immune cells to tumor-draining lymph nodes (tdLN). We isolated these lymph nodes and analyzed migratory and resident DCs as well as monocytes by flow cytometry. Migratory DCs, monocytes, and neutrophils can transport antigens to the draining LN in a Ccr7-dependent manner [3, 14, 15]. Resident DCs can receive those antigens through synaptic transfer [5].

Analysis of lymph nodes harvested from mice injected with B16-ZsGreen tumor cells revealed FP signal in both, migratory and resident DCs, as well as in monocytes. Almost 90% of monocytes, 10–20% of cDC1s, and up to 50% of cDC2s had detectable ZsGreen inside them (Fig 6 and S4 Fig). Other FPs showed only a limited signal suggesting that proteins could have already been degraded on their way to the draining lymph node, perhaps due to shuttling of

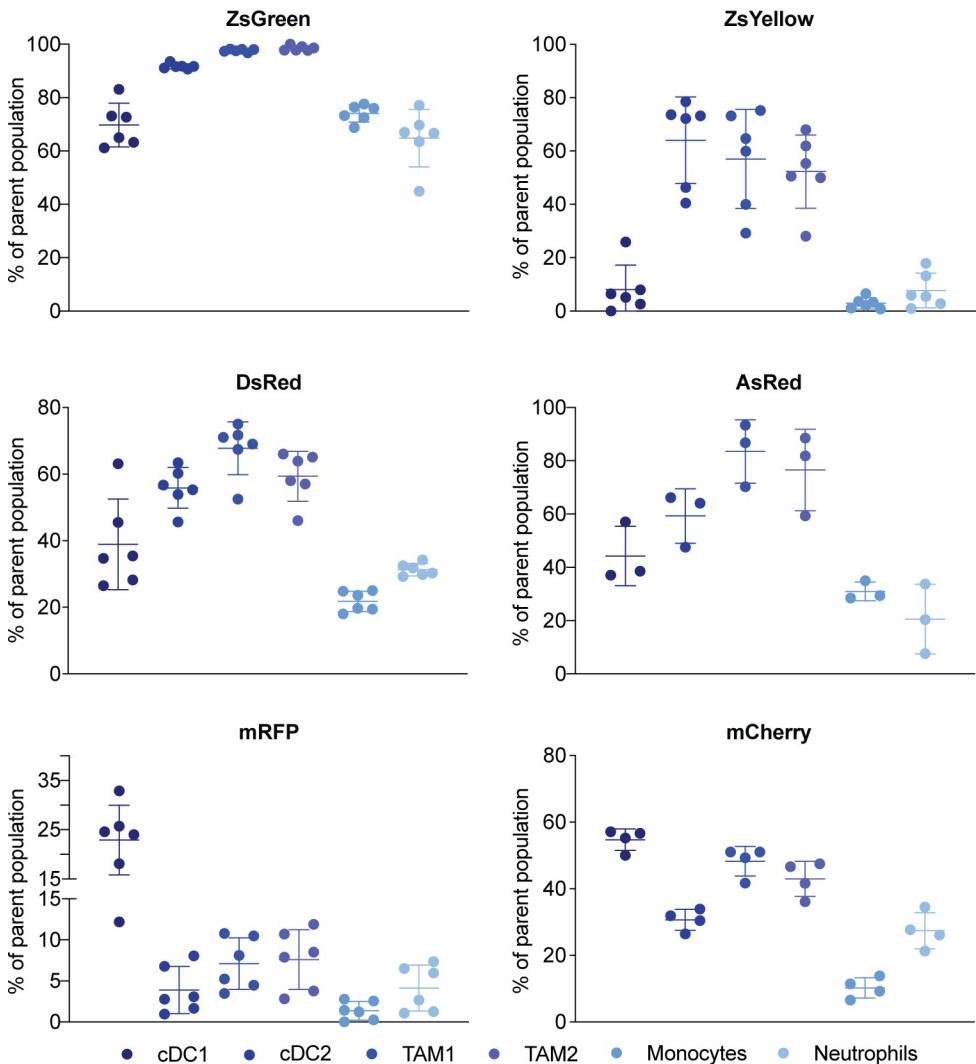

**Fig 4. TAA uptake by tumor-infiltrating immune cells identifies different patterns for different fluorophores.**
Accumulation of respective fluorescent proteins in immune cells, purified from the tumor microenvironment. A non-fluorescent control (B16-F10) was used to set the threshold for FP positivity. One representative example of three independent experiments is shown.

the FPs into a different intracellular environment or organelle. Percentages of ZsYellow and DsRed positive immune cells in respective tdLNs where with up to 20% highest in cDC2s and monocytes, but severly lower than what has been seen for ZsGreen expressing tumors. Fluorescent signal in cDC1s was barely detectable. No signal could be detected in tdLN from mice injected with B16 tumors expressing AsRed, mRFP and mCherry. Interestingly, previous reports showed that mCherry signal could be detected in tdLNs [3]. This difference might be due to different tumor models, different tumor sizes at harvest, and/or the mixture of wild type and mCherry expressing cells in our cell line (Fig 1B).

In summary, ZsGreen was the only FP of the diverse groups of FPs tested that could be tracked efficiently to the tdLN. ZsYellow and DsRed showed significantly less FP signal in immune cells of the tdLN, but could still be considered as potential tools to analyze TAA transfer between immune cells in tdLNs.

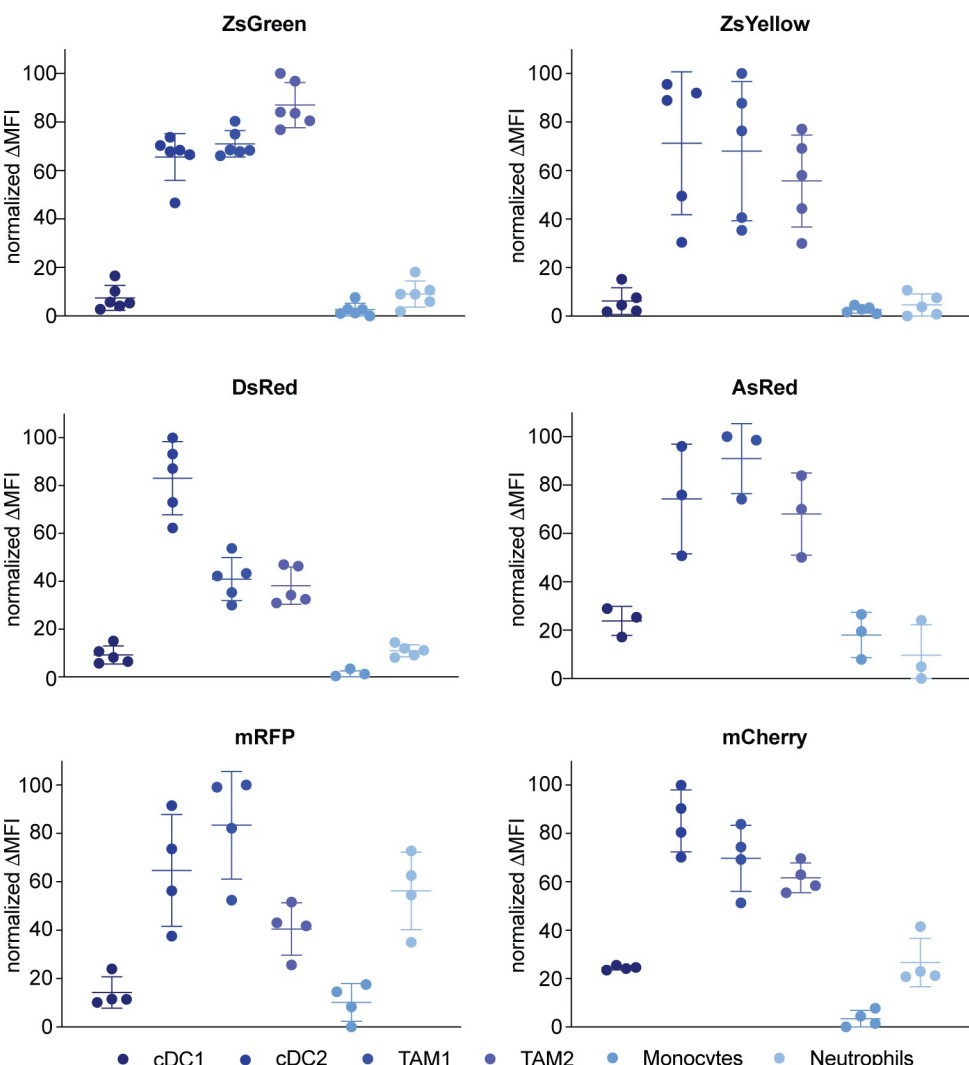

**Fig 5. cDC2s and TAMs contain the most fluorescent TAAs per individual cell.** Immune cells were purified from the TME of B16 mice bearing B16 tumors with the indicated fluorophors and analyzed for relative antigen uptake. Data was normalized to the maximum delta MFI of each FP for every independent experiment.

## Conclusion

In summary, we tested six FPs: ZsGreen, ZsYellow, AsRed, DsRed, mRFP, and mCherry using an antigen uptake and transfer model. Despite comparable *in vitro* stabilities, those FPs showed vastly different loading patterns into tumor-infiltrating immune cells. ZsGreen is the only FP that can be efficiently tracked in lymph node resident DCs and remains the FP to be considered best for use in antigen uptake and transfer assays. We identified ZsYellow and DsRed as additional useful tools for these assays, however, tracking was not as efficient and consistent as for ZsGreen and might need optimization. Additionally, we identified two distinct patterns of FP-antigen loading into tumor-infiltrating immune cells. The 'classical' pattern showed preferential uptake of FPs by cDC2s and TAMs, whereas the 'non-canonical' pattern revealed highest antigen signal in tumor-infiltrating cDC1s. The underlying mechanisms for these distinct antigen uptake patterns remain elusive. Our data suggests that analyses

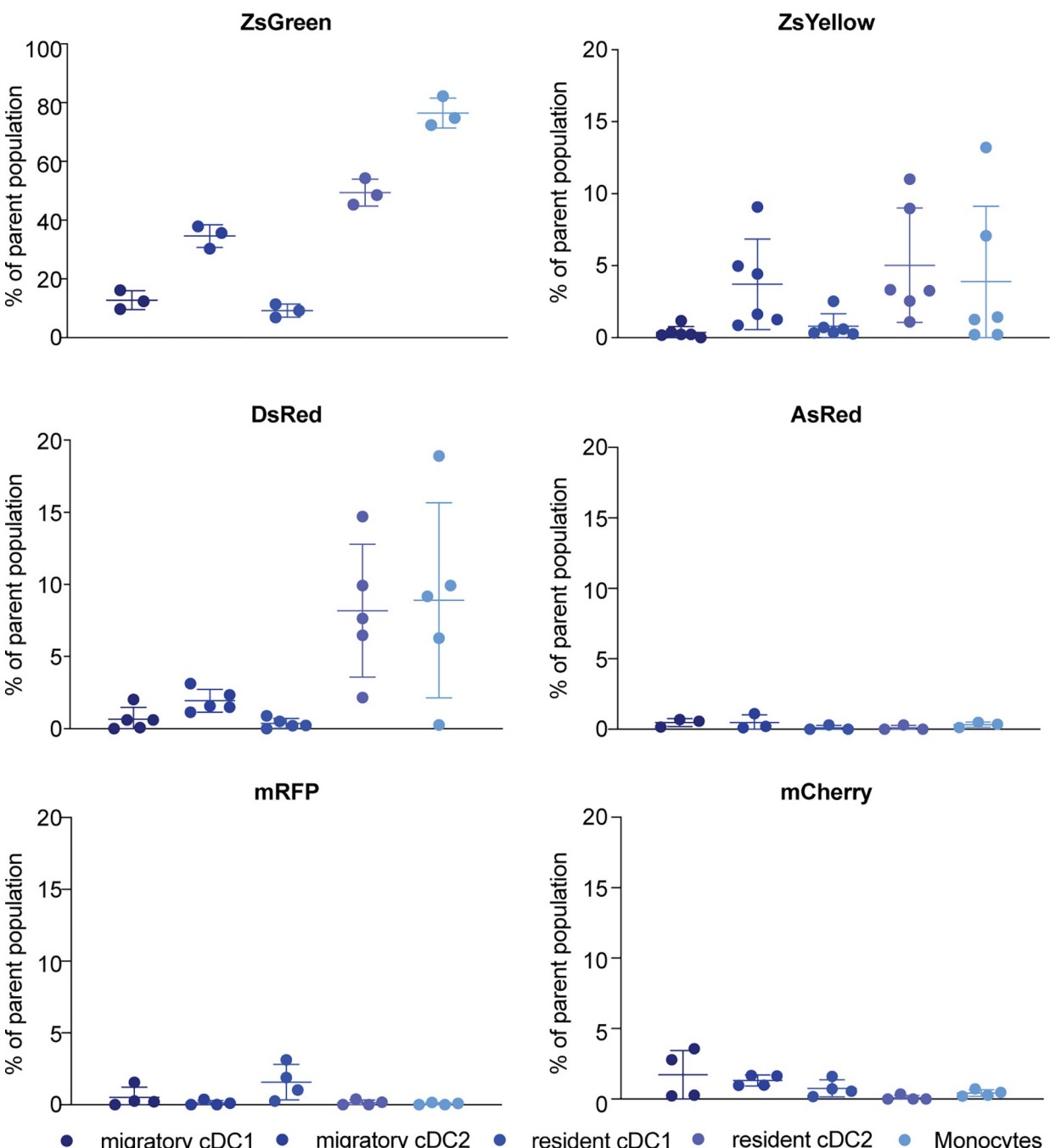

**Fig 6. Limited detection of other non-ZsGreen fluorescent proteins within immune cells of tumor draining lymph nodes.** Tumor draining (inguinal and axillary) lymph nodes were harvested, processed, and analyzed through flow cytometry. Non-fluorescent control (B16-F10) was used to set the threshold for FP positivity. One representative example of three independent experiments is shown.

of antigen uptake and transfer using different FPs should be carried out only after careful consideration of FP uptake patterns. Potential differences might arise as a result of the nature of the chosen FPs.

## Supporting information

**S1 Fig. Similar subcellular localization of fluorescent proteins in FP-expressing B16-F10 cell lines.** Generated B16 melanoma cells with expression of fluorescent protein ZsGreen (**A**), ZsYellow (**B**), AsRed, DsRed, mRFP, and mCherry (all **C**) respectively, overlayed with their bright field images. Images of B16-F10 cell lines were taken to set background fluorescence. (TIF)

**S2 Fig. Gating scheme for flow cytometry analysis of tumor and lymph node samples. A)** Cells collected from tumors were analyzed by flow cytometry using the shown gating methods. Percentages of FP positive cells were analyzed in cDC1, cDC2, TAM1, TAM2, monocytes and neutrophils. **B)** Lymph node cells were analyzed using these gating methods. The dump- populations are B220, CD 90.2, NK 1.1 lineage marker negative cells. Percentage of FP positive cells were analyzed in migratory cDC1, migratory cDC2, resident cDC1, resident cDC2 and monocytes. (TIF)

**S3 Fig. Tumor cell population distributions.** All tumor cells collected were analyzed for their respective immune cell markers. The cell count for each type of immune cell is then plotted as percentages over the total number of live CD45+ cells. (TIF)

**S4 Fig. Lymph node cell population distributions.** All lymph node cells collected were analyzed for their respective immune cell markers. The cell count for each type of immune cell is then plotted as percentages over the total number of live CD45+ cells. (TIF)

## Author Contributions

**Conceptualization:** Rulan Yi, Edward W. Roberts, Matthew F. Krummel, Nina Kathrin Serwas.

**Data curation:** Rulan Yi, Emily Chen, Edward W. Roberts, Nina Kathrin Serwas.

**Funding acquisition:** Matthew F. Krummel, Nina Kathrin Serwas.

**Supervision:** Edward W. Roberts, Matthew F. Krummel, Nina Kathrin Serwas.

**Validation:** Nina Kathrin Serwas.

**Visualization:** Rulan Yi, Nina Kathrin Serwas.

**Writing – original draft:** Rulan Yi, Nina Kathrin Serwas.

**Writing – review & editing:** Edward W. Roberts, Matthew F. Krummel, Nina Kathrin Serwas.

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
