## [Decision Letter · Decision Letter 0]

18 Mar 2022

PONE-D-21-35588Impact of Protein Identity on Tumor-Associated Antigen Uptake into Infiltrating Immune Cells: A Comparison of Different Fluorescent Proteins as Model AntigensPLOS ONE

Dear Dr. Serwas,

Thank you for submitting your manuscript to PLOS ONE. After careful consideration, we feel that it has merit but does not fully meet PLOS ONE’s publication criteria as it currently stands. Therefore, we invite you to submit a revised version of the manuscript that addresses the points raised during the review process.

First, please accept my apologies for the delay, as it was very difficult finding appropriate reviewers.  As you can see, Reviewer #1 was very supportive while reviewer #2 was not as positive. I suggest you address the comments of reviewer #2 and I will happily consider publication of a revised manuscript. If you feel additional experiments are not warranted, please be as comprehensive as possible in describing your reasoning in your response.   Please ensure that your decision is justified on PLOS ONE’s publication criteria and not, for example, on novelty or perceived impact.

We look forward to receiving your revised manuscript.

Kind regards,

Joseph J Barchi

Academic Editor

PLOS ONE

Journal Requirements:

Reviewers' comments:

Reviewer's Responses to Questions

**Comments to the Author**

1. Is the manuscript technically sound, and do the data support the conclusions?

Reviewer #1: Yes

Reviewer #2: Partly

2. Has the statistical analysis been performed appropriately and rigorously? 

Reviewer #1: Yes

Reviewer #2: Yes

3. Have the authors made all data underlying the findings in their manuscript fully available?

Reviewer #1: Yes

Reviewer #2: Yes

4. Is the manuscript presented in an intelligible fashion and written in standard English?

Reviewer #1: Yes

Reviewer #2: Yes

5. Review Comments to the Author

Reviewer #1: Summary: In their report, Serwas et al. conduct a systematic head-on comparison of six fluorescent proteins with a variety of optical features, for their ability to be employed for tumor-assisted antigen uptake studies. While a range of prior reports have independently evaluated these features and unanimously suggested the benefits of ZsGreen, their head-on comparison is unprecedented. This study is important in assisting the choice of the right fluorescent protein depending on the type of study being conducted. The authors report the generation of cell lines with the six fluorescent genes, followed by the evaluation of their in vitro stability. Eventually the genetically modified cell lines are introduced in vivo, and the tumors and the tumor draining lymph nodes are harvested and evaluated for the immune cell population using flow cytometry.

Review:

The paper is scientifically sound and thoughtfully presented. The observation of two different patterns of fluorescent protein-antigen loading in the cells infiltrating the tumors is interesting while the mechanism remains unknown. Notably, all the fluorescent proteins show a similarity in the relative excess of the tumor-associated antigens in cDC2, and tumor associated macrophages. The experimental work is well done and presented clearly, with the authors highlighting the limitations and proposing the potential alternatives. This novel study opens interesting perspectives for the choice of fluorescent proteins for studying specific antigen uptake pattern. The authors do a good job on reporting the experimental outcomes and logistically address the pros/cons of each evaluated protein. The reviewer particularly appreciates the discussion of the methodology used for gating in flow cytometry section. This is a good practice and is hopefully normalized. The referencing is appropriate and well-formatted. I recommend its publication in PLOS One in its current form.

Minor comments:

1. The authors need to confirm if the mode of transfection was through retroviral or lentiviral. There seems to be a discrepancy in the text and figure legend.

2. The authors are encouraged to fix the minor typos in the document.

3. The authors allude to the possibility of certain fluorescent proteins degrading prior to being up taken in the lymph nodes. However, the reviewer wonders if this could this also be seen as a function of the changes in fluorescence in the variety of environments they are exposed to.

Reviewer #2: In this manuscript, authors systematically compared six different fluorescent proteins, namely ZsGreen, ZsYellow, DsRed, AsRed, mCherry, and mRFP. Stability of fluorescent signal was assessed in vitro over time, across different pH environments, and in vivo through FP antigen uptake and transfer to immune cells isolated from tumors and tumor-draining LNs. The mechanism of this process is unclear, which made the paper is not comprehensive and convincing. It is reasonable to conduct computational simulation and simply structure alignment to investigate the antigen uptake of those FPs.

6. PLOS authors have the option to publish the peer review history of their article (what does this mean?). If published, this will include your full peer review and any attached files.

Reviewer #1: **Yes: **Siddharth Sai Matikonda

Reviewer #2: **Yes: **Yanting Xing

---

## [Author Response · Author response to Decision Letter 0]

18 May 2022

Thank you all very much for your time and enthusiasm for the data and the conclusions we presented. We find your comments very helpful to provide us a path toward publication in PLOS ONE. Below you will find detailed responses to the reviewer comments. 

We want to mention that we also added two additional references which are not responses to reviewers. Since the initial submission in November 2021, a new paper showed that neutrophils can transport bacterial antigens to the lymph nodes in a Ccr7-dependent manner (Özcan et al., Science Immunology (2022), 9126, 7(68)). We think this piece of information is important as the study that we cite for the transport of tumor antigens to the lymph node did not completely rule out neutrophils as a possible transport vehicle. Hence, we modified the text and included the above-mentioned citation and the original citation showing transport of bacterial antigens to the lymph nodes:

“…Migratory DCs, and monocytes, and neutrophils are canresponsible for the transport of tumor-associated antigens to the draining tdLN in a Ccr7-dependent manner [3,14,15]., whereas r Resident DCs can receive those antigens through synaptic transfer [5].

Reviewer’s comments

Reviewer #1:

1. Is the manuscript technically sound, and do the data support the conclusions?

Reviewer #1: Yes

2. Has the statistical analysis been performed appropriately and rigorously? 

Reviewer #1: Yes

3. Have the authors made all data underlying the findings in their manuscript fully available?

Reviewer #1: Yes

4. Is the manuscript presented in an intelligible fashion and written in standard English?

Reviewer #1: Yes

5. Review Comments to the Author

Reviewer #1: Summary: In their report, Serwas et al. conduct a systematic head-on comparison of six fluorescent proteins with a variety of optical features, for their ability to be employed for tumor-assisted antigen uptake studies. While a range of prior reports have independently evaluated these features and unanimously suggested the benefits of ZsGreen, their head-on comparison is unprecedented. This study is important in assisting the choice of the right fluorescent protein depending on the type of study being conducted. The authors report the generation of cell lines with the six fluorescent genes, followed by the evaluation of their in vitro stability. Eventually the genetically modified cell lines are introduced in vivo, and the tumors and the tumor draining lymph nodes are harvested and evaluated for the immune cell population using flow cytometry.

Review:

The paper is scientifically sound and thoughtfully presented. The observation of two different patterns of fluorescent protein-antigen loading in the cells infiltrating the tumors is interesting while the mechanism remains unknown. Notably, all the fluorescent proteins show a similarity in the relative excess of the tumor-associated antigens in cDC2, and tumor associated macrophages. The experimental work is well done and presented clearly, with the authors highlighting the limitations and proposing the potential alternatives. This novel study opens interesting perspectives for the choice of fluorescent proteins for studying specific antigen uptake pattern. The authors do a good job on reporting the experimental outcomes and logistically address the pros/cons of each evaluated protein. The reviewer particularly appreciates the discussion of the methodology used for gating in flow cytometry section. This is a good practice and is hopefully normalized. The referencing is appropriate and well-formatted. I recommend its publication in PLOS One in its current form.

We thank the reviewer for the positive comments on our manuscript.

Minor comments:

1. The authors need to confirm if the mode of transfection was through retroviral or lentiviral. There seems to be a discrepancy in the text and figure legend.

Thank you for noticing this. The mode of transfection was indeed retroviral, and we changed that accordingly. We also took the chance to assess the correct use of the terms transduction and transfection and correct the language were needed.

2. The authors are encouraged to fix the minor typos in the document.

We have had multiple proof-reading sessions and believe that we have found and corrected all remaining typos.

3. The authors allude to the possibility of certain fluorescent proteins degrading prior to being up taken in the lymph nodes. However, the reviewer wonders if this could also be seen as a function of the changes in fluorescence in the variety of environments they are exposed to.

In theory, the different fluorescent proteins (FPs) should all experience a similar intracellular environment. However, our data shows that there are substantial differences of protein uptake and maintenance dependent on protein identity. Thus, it might indeed be possible that the FPs are shuttled into distinct organelles, and hence, experience a different intracellular environment. We included a short sentence referring to that possibility in our manuscript:

“…Other FPs showed only a limited signal suggesting that proteins could have already been degraded on their way to the draining lymph node, perhaps due to shuttling of the FPs into a different intracellular environment or organelle…” 

Reviewer #2:

1. Is the manuscript technically sound, and do the data support the conclusions?

Reviewer #2: Partly

2. Has the statistical analysis been performed appropriately and rigorously? 

Reviewer #2: Yes

3. Have the authors made all data underlying the findings in their manuscript fully available?

Reviewer #2: Yes

4. Is the manuscript presented in an intelligible fashion and written in standard English?

Reviewer #2: Yes

5. Review Comments to the Author

Reviewer #2: In this manuscript, authors systematically compared six different fluorescent proteins, namely ZsGreen, ZsYellow, DsRed, AsRed, mCherry, and mRFP. Stability of fluorescent signal was assessed in vitro over time, across different pH environments, and in vivo through FP antigen uptake and transfer to immune cells isolated from tumors and tumor draining LNs. The mechanism of this process is unclear, which made the paper is not comprehensive and convincing. It is reasonable to conduct computational simulation and simply structure alignment to investigate the antigen uptake of those FPs.

We thank the reviewer for critically assessing the quality of our study. We agree with the reviewer that the underlying mechanisms for transfer of FPs, or more generally speaking tumor-associated antigens, to immune cells is not well understood. Previous work suggest that myeloid cells engulf dying tumor cells and ingest them before preparing containing proteins for presentation on their immune cells (Broz et al., Cancer Cell (2014), 638-52, 26(5); Salmon et al., Immunity (2016), 924-38, 44(4); Maier et al., Nature (2020), 257-62, 580(7802)). However, there are a multitude of ways how particles can be ingested into cells (Kumari et al., (2010), 256-75, 20(3)).

The exact consequences of the mode of transfer are not well understood but might be potentially important to determine the elicited immune response. As such we demonstrated in previous work that transferred tumor antigen can be traded among immune cells in the draining lymph nodes (Ruhland et al., Cancer Cell (2020), 786-799, 37(6)). To fully elucidate the underlying mechanism of transfer, and to define consequences of antigen transfer, it is tremendously important to choose the right model antigen. We believe our paper will assist researchers to make the right choice of fluorescent protein. Our findings are important to consider when choosing model antigens for the analysis of antigen uptake and transfer.

The reviewer requests us to undertake a crystallography study:

For us, it is not clear how simple structure alignment and computational simulations should elucidate the mechanism of differential antigen transfer. Only 3 of the chosen fluorescent proteins have available 3D structures on the protein data bank, namely ZsGreen, DsRed and mCherry. The protein structure of fluorescent proteins is quite conserved, so it is not surprising that simple structure alignments with the Matchmaker tool of UCSF ChimeraX (Pettersen et al., Protein Science (2021), 70-82, 30 (1)) result in a substantial overlap of the structures (see Reviewer Figure 1 in the Response to Reviewer document). However, the entire understanding of what will make these structures more or less labile is complex and, we feel, well beyond the goal of this publication which is intended to highlight important differences.

At the highest level of abstraction, the main difference between these chosen proteins is the tendency to form dimers, tetramers, or remain as monomers. It is not clear whether these changes result in different antigen transfer, uptake, and/or maintenance, but could be a possibility. We added a sentence, and a corresponding reference, reflecting the different tendencies for multimerization of the proteins into our manuscript:

“…It remains unclear what causes the increased percentage of FP-positive tumor infiltrating cDC1s in mRFP and mCherry-expressing tumors. Numbers of cDC1s are not different between tumors expressing FPs irrespectively of whether they cause ‘classical’ or ‘non-canonical’ antigen uptake patterns (S3 Fig). Of note, whereas ZsGreen, ZsYellow, AsRed, and DsRed are proteins directly isolated from distinct organism, mCherry and mRFP have been designed to better serve as tools for molecular biology. Changes aimed to enhance brightness and stability, and to prevent dimerization [10,13]. ...”

---

## [Decision Letter · Decision Letter 1]

28 Jul 2022

Impact of Protein Identity on Tumor-Associated Antigen Uptake into Infiltrating Immune Cells: A Comparison of Different Fluorescent Proteins as Model Antigens

PONE-D-21-35588R1

Dear Dr. Serwas,

We’re pleased to inform you that your manuscript has been judged scientifically suitable for publication and will be formally accepted for publication once it meets all outstanding technical requirements.

Kind regards,

Joseph J Barchi

Academic Editor

PLOS ONE

Additional Editor Comments (optional):

Reviewers' comments:

Reviewer's Responses to Questions

**Comments to the Author**

1. If the authors have adequately addressed your comments raised in a previous round of review and you feel that this manuscript is now acceptable for publication, you may indicate that here to bypass the “Comments to the Author” section, enter your conflict of interest statement in the “Confidential to Editor” section, and submit your "Accept" recommendation.

Reviewer #2: All comments have been addressed

2. Is the manuscript technically sound, and do the data support the conclusions?

Reviewer #2: Yes

3. Has the statistical analysis been performed appropriately and rigorously? 

Reviewer #2: Yes

4. Have the authors made all data underlying the findings in their manuscript fully available?

Reviewer #2: Yes

5. Is the manuscript presented in an intelligible fashion and written in standard English?

Reviewer #2: (No Response)

6. Review Comments to the Author

Reviewer #2: (No Response)

7. PLOS authors have the option to publish the peer review history of their article (what does this mean?). If published, this will include your full peer review and any attached files.

Reviewer #2: **Yes: **Yanting Xing

---

## [Editor Report · Acceptance letter]

9 Aug 2022

PONE-D-21-35588R1 

Impact of Protein Identity on Tumor-Associated Antigen Uptake into Infiltrating Immune Cells: A Comparison of Different Fluorescent Proteins as Model Antigens 

Dear Dr. Serwas:

I'm pleased to inform you that your manuscript has been deemed suitable for publication in PLOS ONE. Congratulations! Your manuscript is now with our production department. 

Kind regards, 

on behalf of

Dr. Joseph J Barchi 

Academic Editor

PLOS ONE